# Telomere Biology and Human Phenotype

**DOI:** 10.3390/cells8010073

**Published:** 2019-01-19

**Authors:** Kara J. Turner, Vimal Vasu, Darren K. Griffin

**Affiliations:** 1University of Kent, School of Biosciences, Giles Lane, Canterbury, Kent, CT2-7NJ, UK; k.j.turner-24@kent.ac.uk (K.J.T.); vimal.vasu@nhs.net (V.V.); 2Department of Child Health, East Kent Hospitals University Foundation NHS Trust, William Harvey Hospital, Ashford, Kent, TN24-0LZ, UK

**Keywords:** telomeres, telomere length, aging, senescence

## Abstract

Telomeres are nucleoprotein structures that cap the end of each chromosome arm and function to maintain genome stability. The length of telomeres is known to shorten with each cell division and it is well-established that telomere attrition is related to replicative capacity in vitro. Moreover, telomere loss is also correlated with the process of aging in vivo. In this review, we discuss the mechanisms that lead to telomere shortening and summarise telomere homeostasis in humans throughout a lifetime. In addition, we discuss the available evidence that shows that telomere shortening is related to human aging and the onset of age-related disease.

## 1. Introduction: Structure, Function and Maintenance of the Telomere

Telomeres are nucleoprotein structures found at the end of each chromosome arm that function to maintain genome stability. In all mammals, telomeres are formed of a highly conserved, hexameric (TTAGGG) tandem repeat DNA sequence. This is organised into a looped structure called a T-loop and associated with specialised proteins including, among others, those that make up the Shelterin complex [1,2,3]. The looped structure (Figure 1) is formed via nucleolytic activity at the extreme termini of telomeric DNA to produce a single stranded G-rich overhang. This loops back and invades the double stranded telomere tract [4,5], ensuring that loose DNA ends are housed internally within the nucleoprotein structure.

There are many proteins associated with the telomere that, combined, make up the telosome. Some are involved in DNA damage response mechanisms, for example DNA protein kinase (DNA-PK), p53, polyadenosine diphosphate ribose polymerase (PARP), Tankyrase 1 and 2, Excision repair cross-complementing associated with xeroderma pigmentosum group F (ERCC/XPF) radiation 51 (RAD51), werner (WRN) and bloom (BLM) [6,7]. Others are involved in nuclear organization such as lamin associated proteins (LAPs) [8,9] and silent information regulator (Sir) proteins, which are also involved in epistatic control of telomere length [10]. The presence and action of these proteins at the telomere sequence is largely governed by proteins that make up the Shelterin complex. This complex is made up of a collection of six specialized proteins that associate with the telomere structure to form a fully functional capping structure. These proteins and their characteristics are outlined in further detail in Table 1.

Interactions between members of the Shelterin complex and the telomere DNA sequence stabilise the telomere structure and regulate access of proteins involved in DNA repair and lengthening [3,11,13]. Collectively therefore, this specialised nucleoprotein structure functions to form a cap at the chromosome ends, serving two main functions: Firstly it shields the ends of chromosome arms from inappropriate DNA repair mechanisms, which might otherwise recognise loose DNA strands as a double strand break and result in chromosome fusion events. Secondly, this capping structure prevents the degradation of genes near the end of chromosome arms as a result of incomplete DNA replication.

During DNA replication, DNA polymerase may only synthesise new DNA in the 5′ to 3′ direction with the aid of RNA primers that are formed by the enzyme primase. These primers anneal to the template strands, providing a free 3′ OH group for the addition of free nucleotides. Since the action of polymerase proceeds in the same direction as the progression of the replication fork, the synthesis of new DNA in the 5′ to 3′ direction requires only one primer and is continuous [14]. However, synthesis of a new 3′ to 5′ DNA strand progresses against the direction of the replication fork. Therefore synthesis of this new strand requires the annealing of multiple RNA primers which are elongated into short Okazaki fragments and subsequently ligated to form the new DNA strand [15]. Consequently, a length of DNA at least the size of the RNA primer is lost at the 5′ end of the lagging strand when the final RNA primer is removed following replication [16]. In reality however, a much larger amount of DNA is lost following replication as a consequence of priming failure [14] and as a consequence of the complex nature of the telomere structure itself. The association of telomere sequences with the Shelterin complex leads to replication fork stalling [17] and the presence of G-quadruplexes, which are an essential feature in the formation of the t-loop, results in replication fork slippage [5]. Furthermore, the synthesis of DNA via polymerase action produces blunt ends. In order to form a T-loop, nucleolytic activity is required to produce a single stranded overhang (Figure 1). Thus, there are several mechanisms that result in telomere attrition following replication in proliferating cells and it is thought that the presence of the telomere sequence acts as a ‘buffering system’ to prevent the loss of crucial DNA. Eventually however, this buffering system is lost and it is widely accepted that, as a consequence of this, the cell loses its ability to proliferate and reaches its so-called replicative capacity.

Once telomeres have shortened beyond a critical level, the proteins that form the Shelterin complex are unable to associate with the telomeric sequence and can no longer perform their role in capping the end of the chromosome. Therefore, a major limiting factor in the function of the telomere is its length. This represents a problem in cells that are highly mitotically active, such as stem cells that differentiate in order to generate new tissue or replace damaged cells. In these cells, it is important that telomere length is maintained to ensure prolonged replicative capacity and this is achieved via two primary mechanisms: The action of a specialised enzyme called telomerase or homologous recombination mediated alternative lengthening of telomeres (ALT). Telomerase is a ribonucleoprotein complex, made up of a telomerase reverse transcriptase (TERT) catalytic subunit that synthesises new telomeric repeats by copying its telomerase RNA component (TERC) [18]. In most cell types, its action is inhibited by the competitive binding of telomeric repeat containing RNA (TERRA) (which is transcribed from sub-telomeric and telomeric DNA sequences) to TERC and via contact with TERT [19,20]. Several cell types including stem cells are able to express telomerase however and in cells that do not (or where telomerase is repressed) homologous recombination based ALT may extend telomere length. In this scenario it is thought that the 5′ overhang belonging to the telomere of one chromatid may invade the T-loop of the homologous chromatid. This structure resembles a replication fork that is recognised by DNA polymerase and is subsequently extended [21].

Despite the presence of these elongation mechanisms, telomere shortening is still observed following proliferation in most stem cells (with the exception of embryonic stem cells). This is because telomere elongation mechanisms are increasingly reduced during differentiation and therefore they become insufficient to completely eradicate telomere loss. In the majority of differentiated somatic cells (except for lymphocytes), telomerase is not expressed at all and therefore in these cells, telomere length declines with each division [22].

## 2. Telomere Length and Replicative Capacity

The finite replicative capacity of somatic cells has been recognised since Leonard Hayflick’s work in the 1960s [23]. However, its association with the ‘end replication problem’ was not discovered until the 1970s and it was a further twenty years before telomere shortening was formally associated with passage number and replicative capacity in vitro [24,25,26]. Since then, an array of studies have shown that telomere length is correlated with donor age in a variety of somatic cell types, generally declining as a function of chronological age [27,28,29,30,31,32,33,34].

Furthermore, the contribution of telomere attrition to the process of aging has been well characterised and it is now generally accepted that when telomere length becomes critically short, the ability to support the Shelterin complex is lost. In turn, the inhibitory action of the Shelterin complex on DNA damage response pathways is released and the cell cycle leaves G1 and enters G0 [35]. This is initiated by the ataxia telangiectasia mutated (ATM) or the ataxia telangiectasia and radiation 3 (RAD3) related protein (ATR) pathway. Both of these lead to the phosphorylation of p53, expression of p21 and inhibition of cyclin dependent kinases that would otherwise enable progression through the cell cycle [36,37]. Upon leaving the cell cycle, the cell either enters senescence (defined as the irreversible cessation of division) or apoptosis, (defined as programmed cell death). The exact mechanisms that dictate the committal to senescence or apoptosis are poorly understood [38] and the characteristics of each scenario are quite different. However, the outcome (inability of the cell to continue division) is the same.

The consequence of an accumulation of senescent cells is two-fold: Firstly, senescence leads to a reduction in the number of mitotically active cells in a given tissue, limiting the potential for growth and repair. Secondly their accumulation results in the release of proteases, growth factors and inflammatory cytokines which act on non-senescent neighbouring cells. Normally, this initiates clearance of senescent cells by the immune system. However, as the immune system ages its ability to clear senescent cells becomes impaired [37,38]. Ultimately therefore, it is thought that the accumulation of senescent cells as a result of telomere attrition drives the process of tissue and organismal aging. In order to test this hypothesis, a multitude of studies have measured telomere length in vivo in relation to chronological age [30,31,32,34]. Moreover, a variety of studies have investigated the relationship between telomere length and age-related disease [39,40,41].

## 3. Telomere Homeostasis Throughout a Life-Time

As male and female gametes fuse during the process of fertilisation, telomere length must be reset in order for the offspring to have sufficient telomeric reserve to develop and fulfil a healthy lifespan. Following fertilisation, telomere length declines in the cleavage stage embryo in comparison to the oocyte and declines further still at the morula stage [42] in line with a decline in telomerase activity during this time [43]. At the blastocyst stage however, telomerase activity is markedly increased and telomeres are lengthened [42,43]. At present, information regarding telomere dynamics in the earliest weeks of gestation are lacking except for a few notable observations. Cheng and colleagues showed that, between weeks six and seven of gestation, telomere length rapidly declines, however this is slowed between weeks eight and eleven and then remains constant thereafter (that is, telomere length in foetuses above eleven weeks’ gestation was not different to that of full term babies) [44]. This finding is supported by others whom concluded that telomere length is not associated with gestational age in foetuses between 15 and 19 weeks’ gestation [45] and that, although telomere length may fluctuate between 23 and 36 weeks’ gestation, overall telomere length is either increased or no detectable change is observed [46]. Furthermore, telomere length appears to be synchronous among different tissues during foetal life [45] and is maintained by telomerase activity in utero [44,47,48].

At the time of birth, many have shown that telomere length is highly variable [49,50,51,52], a finding consistent with the highly variable telomere length observed in human embryos [42] and foetuses [45,46]. Furthermore, telomere length in newborns appears to be associated with parental telomere lengths, though controversy exists as to the relative influence of the mother [53,54] and the father [55,56]. Interestingly, telomere length appears to maintain synchrony between tissues at the time of birth and telomere length appears similar between male and female babies [49,52]. This is in contrast to observations in adults which indicate that while telomere lengths in different tissues may be correlated, they are highly variable [57,58] and that telomere length is longer in women than in men [40,59,60].

Following birth, telomere length begins to decline within the first weeks of postnatal life and interestingly, this pattern holds true for infants born preterm, indicating that foetal telomere length maintenance is specific to life in utero [50,51]. Moreover, it has been shown that the rate of telomere shortening is most pronounced in the early years of life following birth and that this rate of attrition declines in young adulthood and reduces further still in older individuals [29,30,31,32,34]. Despite the relatively earlier onset of telomere shortening in preterm infants in comparison to term born infants overall telomere length does not appear to be significantly shortened at term equivalent age [49] or in early childhood [61]. Interestingly however, a number of studies have highlighted that telomere attrition rate in adults is most prominent in those with higher telomere length at baseline [59,62,63].

### Telomere Length in Relation to Demographic and Lifestyle Factors

Evidence to support the hypothesis that telomere length is inversely correlated with chronological age is well-documented within the literature. However, it is important to note that in many studies the data appears to be somewhat scattered and the correlations between telomere length and chronological age may be less well-established than previously thought. In addition, telomere length has been shown to be highly variable among individuals of the same or similar age in all age ranges assessed. Alongside chronological aging, available evidence (summarised in Table 2) supports the notion that a wealth of genetic and environmental factors may modulate telomere length.

## 4. Telomere Length and Biological Aging

The high inter-individual variability of telomere length means that, although it is convincingly associated with the finite replicative capacity of cells in vitro, its relationship with the process of biological aging in vivo is more difficult to unpick. Aging can be defined as the gradual decline in normal tissue and organ function over time as a consequence of an accumulation of senescent cells and a decline in the regenerative potential of stem cells [84,85]. This decline in tissue function can be thought of as normal ‘wear and tear’ that occurs over time and may be the sole cause or a contributing factor to the development of age-related diseases in combination with other inducers of cell senescence [84]. Such inducers might include endogenous factors; for example mitochondrial dysfunction [86,87] and inflammation [88] or exogenous factors; such as cigarette smoking, high fat diet, chemotherapy, radiation and other environmental or lifestyle factors [84,89]. These factors are strongly linked with the production of reactive oxygen species (ROS), which are known to induce cellular senescence and moreover [89], it is thought that the G-rich telomere repeat sequence is particularly susceptible to oxidative damage [90]. Since telomere attrition is an initiator of cell senescence in vitro, a variety of studies have sought to investigate the relationship between telomere shortening and age-related diseases. These commonly occur during the process of normal aging but in rare examples, may occur in the context of premature aging disorders.

### 4.1. Telomere Biology and Premature Aging Disorders

Premature aging disorders result in characteristic symptoms normally associated with old age, such as hair greying, hair loss, neurological degeneration, loss of subcutaneous fat, skin atrophy and cancers. In all cases, genetic mutations lead to dysfunctional proteins that are involved in DNA damage response and/or DNA damage repair pathways, telosome structure or telomere length regulation. As a result, depending on the affected pathway, telomere length may be shortened in these individuals or the rate of telomere attrition may be accelerated in comparison to age matched individuals. In recent decades, a number of studies have investigated telomere biology in relation to premature aging disorders and the key findings from these are highlighted in Table 3. However, while these studies have provided some valuable insights into the role of telomere biology in the pathogenesis of age-related disease, many aspects of premature aging disorders are not observed as part of normal aging and therefore it is difficult to apply these findings to aging in the general population [40].

### 4.2. Telomere Length in Age-Related Cardiometabolic and Neurological Disorders

Several studies have shown that shortened telomeres are associated with cardiovascular disease [118,119,120], including atherosclerosis [121,122,123], hypertension [124], vascular dementia [125] and coronary heart disease [126]. Moreover, in many cases telomere length has been identified as an indicator of the severity of such conditions [60,127] and has been associated with risk of stroke, heart attack and mortality [127,128]. However, methodological issues, particularly in relation to adjustment for important confounders (e.g., age, gender and ethnicity) mean than drawing robust conclusions from the data is difficult. Furthermore, others have noted that while telomere length itself may not be a risk factor for mortality associated with cardiovascular disease, the rate of telomere attrition is [129].

Type II diabetes is another disease that is recognised as part of the aging phenotype and is one of the most common chronic diseases in the world. Again, this disease shows conflicting results across different studies that have assessed its relationship with telomere length. Initial studies showed that short telomeres are associated with type II diabetes [130,131]; however in prospective studies this association was not always replicated. While some analyses demonstrate shortened telomere length as a risk for type II diabetes, others report no such link [132,133]. A recent meta-analysis argued that this conflicting information may be due to small sample sizes in previous studies and therefore the authors pooled data from a large number of studies, concluding that shortened telomeres are associated with type II diabetes [134]. The authors point out however, that the strength of association in sub-group analysis was influenced by age and that other studies identify additional influencers of telomere length such as gender and ethnicity. Information on these factors was insufficient for such analysis in their own study and therefore further research is warranted [134].

Similar conclusions can be drawn from studies that have investigated telomere length in Alzheimer’s disease patients and in those suffering with dementia. Several studies show that affected individuals possess shortened leukocyte telomere lengths in comparison to age matched controls [135,136,137,138,139], however the data that associates severity of disease with telomere length is less clear-cut. While some show that telomere lengths of dementia and Alzheimer’s disease patients are correlated with disease status [136,138,139] others have not [140,141,142].

In addition, studies that have investigated telomere length in Parkinson’s disease patients show similar results. Although telomere length has been shown to be reduced compared to controls in one study [143], this was not observed in another [144]. Furthermore, a third study found that individuals with shortest telomere lengths were three fold less likely to develop Parkinson’s disease [145]. That being said, it has been noted that the range of telomere lengths is altered in Parkinson’s disease patients, with lengths of less than 5kb only observed in patients and not controls. Furthermore, in older individuals (above 60 years old), the percentage of short telomeres increased over time in comparison to controls [144].

### 4.3. Telomeres, Tumorigenesis and Cancer

Cancer, which is defined as the uncontrolled growth of abnormal cells, is the leading cause of death worldwide [146] with advancing age being the most significant risk factor [147]. Cancer cells are typically characterised by chaotic genome instability and immortality as a result of an acquired means to circumvent normal replicative barriers. Therefore, the finite replicative capacity of the cell is vital in the prevention of cancer and in this sense, telomere function may also be extended to the protection against cancer. Once telomere degradation has reached a critical level, the cell is committed to pathways that ultimately result in senescence or apoptosis and thus telomeres protect the integrity of the genome. However in the event that this mechanism should fail, tumorigenesis ensues [148].

Tumorigenesis is considered to be a three-step process, in which cancer cells evolve the ability to overcome committal to senescence. First, are the loss of telomeric repeat sequences and/or the loss of the telosome structure, which may occur independently of or as a result of the former [149]. Second is the inappropriate action of non-homologous end joining (NHEJ) or homology direction repair (HDR) machinery, which may recognise uncapped telomeres as DNA breaks. Such action results in the fusion of chromosomes at their ends, generating dicentric chromosomes which are subsequently pulled apart during cell division, creating further breakages. This breakage-bridge-fusion cycle continues during successive cell divisions resulting in duplication of whole chromosomes, aneuploidy, gene amplifications, translocations, inversions and deletions [150]. Such complex chromosome rearrangements add further oncogenic potential via deregulation of oncogenes or altered gene dosage [151,152,153]. This second stage, known as the crisis stage, drives malignant transformation via the ability to evade apoptosis mechanisms [154]. Furthermore, as more successive divisions occur, telomeres shorten further, encouraging more chromosomes to enter breakage-fusion-bridge cycles and an accumulation of genomic instability [150]. Finally, these malignant cells must acquire immortality in order to continue cell division unchecked. In 90% of cancers, this is achieved via the expression of telomerase, which is inactive in normal somatic cells and acts to maintain telomere length in a shortened state [155]. Less commonly, this maintenance is achieved via ALT [156].

With the above in mind, it is widely believed that shortened telomeres are both a protector (when recognised via the appropriate mechanisms) and an initiator (when not recognised) of tumorigenesis. In recent decades, it has also been increasingly recognised that a single short telomere may be sufficient to initiate tumorigenesis and that telomerase action may be tightly controlled, such that it may selectively act upon only the shortest telomeres. This in turn leads to the maintenance of a shortened state, synchronous with other chromosomes within the population of cells rather than elongation beyond the length of other chromosomes. Therefore, in many (but not all) tumours, overall telomere length may be unchanged or remain shortened in comparison to normal neighbouring cells [157,158,159].

Given the association between shortened, dysfunctional telomeres and tumorigenesis it is unsurprising that many researchers have investigated the relationship between telomere length and a variety of specific cancers. In similarity to its relationship with chronological age, the relationship between telomere length and cancer is equally difficult to draw robust conclusions from. The majority of studies report shortened telomere length and some have additionally found that the degree of malignancy and prognosis were also associated with telomere length [160,161]. However this appears to be specific to certain types of cancer. In meta-analyses, telomere length was shown to be shortened in bladder, oesophageal, gastric, head and neck, ovarian and renal cancers [160]. Short telomeres and telomerase mutations are also associated with hepatocarcinoma [162,163]. However, no association with telomere length was observed in endometrial, prostate and skin cancer. Furthermore, an assessment of the association between telomere length in non-Hodgkin’s lymphoma, breast, lung and colorectal cancer proved inconclusive [160]. Moreover, a recent systematic analysis of telomere length in 31 cancers showed that while overall, telomere length was shortened in tumour compared to normal tissues, many tumour types showed telomere elongation in a proportion of samples assessed. In three tumour types (testicular germ cell carcinoma, lower grade glioma and sarcoma), over 50% of samples showed telomere elongation [164]. Similarly, contradictory results are available from a variety of studies that have associated telomere length with the risk of developing cancer. While some studies report that longer telomeres are associated with an increased risk of developing cancer [165,166,167], others argue that shortened telomeres increase cancer risk [160,168,169]. These discrepancies may at least in part, be down to differences in study design. Prospective studies assess telomere length prior to diagnosis and therefore prior to the crisis stage of tumorigenesis when telomeres are longer and inhibit the protective effects of senescence whereas retrospective studies assess telomere length after cancer diagnosis and therefore after the crisis stage of tumorigenesis when telomeres are shorter and genomic instability ensues [165,170].

## 5. Conclusions and Perspectives

Since their discovery in 1939, our knowledge of telomere biology has continued to advance in leaps and bounds. Available data suggest that telomere attrition is associated with cellular senescence, the process of aging and the pathogenesis of many diseases. However, whilst these findings are both interesting and invaluable areas of uncertainty remain. Firstly, it must be recognised that association does not imply causality and the observational nature of telomere studies precludes any causal inferences. Secondly, whether shortening of telomeres observed is cause, effect or both is not well established. In addition, high inter-individual variability in telomere length suggests that many other factors besides chronological age may act as influencers (Table 2). Furthermore, many other factors in addition to telomere shortening may contribute to the process of aging at the cellular, organ and organismal level. For example genetic, epigenetic, environmental and lifestyle factors [171,172] have been implicated in addition to mechanisms that regulate protein homeostasis, nutrient sensing [173] and mitochondrial function [173,174].

It is important to recognise that telomere length is a difficult parameter to measure and therefore technical factors in any study design may also impose significant problems in the interpretation of study results. For example, sample storage conditions and the methodology used to extract DNA and measure telomere length [175,176,177] may have an effect on the results obtained. In the context of studies that have assessed telomere length in relation to the onset of disease, the origin of the sample (e.g., the cell type and whether the sample is specific to the affected tissue or a surrogate sample) [160], the type of controls used (paired samples with normal controls from the same individual or unpaired samples with controls from healthy donors) the timing of the sample (prior to or subsequent to the onset of disease) and the administration of any treatment [165] may have additional effects. In the future therefore, research efforts should employ carefully designed, robust and reproducible methodologies in order to further our understanding of how the complex mechanisms that orchestrate the relationship between telomere biology and the process of aging and disease are interwoven.

## Figures and Tables

**Figure 1 cells-08-00073-f001:**
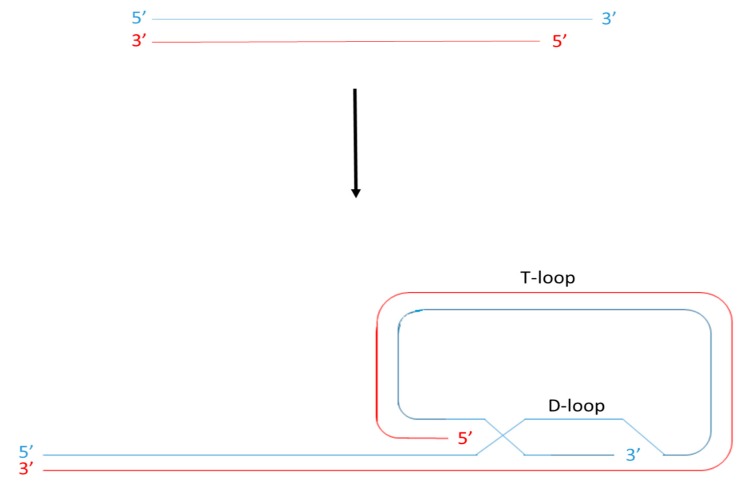
The T-loop and the D-loop. The 3′ end of the G rich strand (blue) protrudes as a single stranded extension of the telomere. This G-strand overhang loops back to form a T-loop and invades the 5′ double stranded telomeric duplex, forming a D-loop.

**Table 1 cells-08-00073-t001:** The Shelterin complex: Characterisation of the proteins that make up the Shelterin complex [3,11,12].

Protein Name	Interactions	Function
Telomere repeat binding factor 1 (TERF1 also known as TRF1)	Direct interaction with double stranded TTAGGG repeats	Regulation of telomere length
Telomere repeat binding factor 2 (TERF2 also known as TRF2)	Direct interaction with double stranded TTAGGG repeats	Stabilisation of the T-loop and regulation of telomere length
TERF1 interacting nuclear factor 2 (TINF2 also known as TIN2)	Associates directly with TERF1, TERF2 and ACD and indirectly with POT1	Tethering of ACD and POT1 to TERF1 and TERF2 and tethering TERF1 to TERF2, which stabilises the association of TERF2 with the telomere. Also regulates telomere length
Protection of Telomeres 1 (POT1)	Direct interaction with single stranded telomere overhang	Inhibition of DNA damage response and regulation of telomere length
Shelterin complex subunit and telomerase recruitment factor (ACD, also known as TPP1)	Interaction with TINF2 and POT1	Enhances POT1 binding to single stranded telomere DNA and regulates telomere length in combination with POT1
TERF2 interacting protein (TERF2IP also known as RAP1)	Associates with TERF2	Telomere length regulation

**Table 2 cells-08-00073-t002:** Demographic factors affecting telomere length: An overview of current findings and references.

Demographic Factors	General Observations	References
Genetic factors	Several twin studies have identified high heritability of telomere length and many specific loci associated with telomere length have been reported.	[64,65,66,67]
Gender	Longer telomeres are found in adult females compared to males. This is thought to be due to higher levels of oestrogen, which confers anti-inflammatory as well as antioxidant properties and is known to promote telomerase expression.	[60,68,69,70]
Ethnicity	Telomeres are slightly longer in white individuals compared to black and Hispanic individuals. However, this difference is often not statistically significant unless also adjusted for other factors such as age, sex, socio-economic background and lifestyle factors (diet and smoking)	[71]
Level of psychosocial stress	Shortened telomeres are associated with high levels of psychosocial stress as a result of increased oxidative stress as well as reduced telomerase activity. Telomere length is also inversely correlated with major depressive disorder due to increased inflammatory factors leading to increased oxidative stress.	[72,73,74]
Level of physical activity	Longer telomeres have been found in those that engage in higher levels of physical activity, which is associated with improved physical and psychological wellbeing. Thus it is possible that the effects of physical activity on telomere length are influenced by a positive effect on physical and mental well-being	[75,76,77]
Obesity	Telomeres are known to be shortened in obese individuals. Obesity is associated with chronic inflammation, increased reactive oxygen species (ROS) production in adipose tissue and evidence of increased systemic oxidative stress. Furthermore, telomere length is correlated with body mass index (BMI), with increased BMI resulting in higher blood volume, stimulating increased proliferation of blood cells and leading to telomere shortening. Interestingly, weight loss is positively correlated with telomere lengthening and those with shortest telomere length at baseline benefit from the most pronounced rate of telomere lengthening following weight loss. A greater adherence to a Mediterranean diet is also associated with longer telomeres.	[78,79,80,81]
Smoking	Telomere length is shorter in smokers and ex-smokers compared to non-smokers and negatively associated with the amount of cigarettes smoked per year.	[78,82]
Alcohol consumption	Telomere length is negatively correlated with the number of alcohol units consumed per day and is shorter in alcohol abusers compared to controls.	[83]

**Table 3 cells-08-00073-t003:** Telomere biology in premature aging disorders: Clinical observations and aberrant telomere observations associated with premature aging disorders.

Premature Aging Disorder	Characteristic Symptoms	Mutations Observed	Effects on Telomere Structure	References
Hutchinson-Gilford Progeria Syndrome	Hair greying and loss, decreased joint mobility, loss of subcutaneous fat and atherosclerosis	Point mutation in the *LMNA* gene encoding prelamin A; a protein involved in nuclear lamina. Mutant *LMNA* induces DNA damage response at the telomere leading to cell senescence	Shortened telomere length	[91,92,93]
Werner Syndrome	Hair greying and loss, skin atrophy, diabetes, osteoporosis, cataracts, arteriosclerosis and neoplasms	Mutation in *WRN* gene located on the P arm of chromosome 8, which encodes the RecQ DNA helicase involved in DNA replication, recombination and repair. Recruitment of WRN by TERF2 is essential for resolution of the telomeric D-loop and synthesis of the telomeric 3′ overhang	Average telomere length is not reduced. However, loss of telomeres on individual sister chromatids is observed leading to chromosome breakage-fusion events, genome instability and cell senescence. The rate of overall telomere attrition is also increased.	[94,95,96,97,98]
Bloom Syndrome	Growth retardation, immunodeficiency, genomic instability cancer and premature menopause	Mutation of BLM; another RecQ helicase associated with TERF2 and involved in DNA replication, recombination and repair	Telomere length is not reduced. However, the rate of telomere shortening is accelerated	[99,100,101,102]
Nijmegen Breakage Syndrome	Chromosomal instability and cancers	Mutation of NSB1, which is involved in DNA repair in association with TERF2	Shortened telomere length	[103,104]
Cockayne Syndrome	Neurological degeneration, hearing loss, retinal degeneration and loss of subcutaneous fat	Mutation in one of five genes including *CSA*, *CSB*, *XPB*, *XPD* and *XPG*. Mutation in *CSB* is implicated in the majority of cases. CSB interacts with TERF2 as well as TERF1 to regulate telomere length maintenance	Shortened telomere length	[105,106,107]
Dyskeratosis Congenita	Abnormal skin pigmentation, nail dystrophy, bone marrow failure and cancer	One of several mutations involving telomerase (an enzyme involved in telomere length maintenance) or proteins that regulate telomerase. In the X-linked recessive form, DKC1 is mutated, which associates with TERC (the RNA component of telomerase). In the autosomal dominant form, TERC is commonly involved; however TIFN2 is mutated in some cases. In autosomal recessive forms, mutations in TERT (the reverse transcriptase component of telomerase), NOP10 and NHP2 are the cause. NHP2 interacts with NOP10, which in turn associates with DKC1 in order to interact with TERC.	Shortened telomere length. Furthermore, shorter telomeres are associated with more severe clinical phenotypes	[108,109,110,111,112]
Ataxia telangiectasia	Neurological deterioration, chromosomal instability and predisposition to cancer	Mutations in ATM, which is located on the q arm of chromosome 11 and is involved in cell cycle progression and DNA repair pathways	Accelerated telomere shortening and chromosome fusion events	[113,114]
Down’s Syndrome	Accelerated aging characteristics such as premature skin wrinkling, greying hair, hypogonadism, hypothyroidism, early menopause and declining immune function. In addition overexpression of amyloid precursor protein (APP) on chromosome 21 leads to Alzheimer’s Disease	Trisomy chromosome 21	Shortened telomere length	[115,116,117]

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
