# Peer review of "Telomere Biology and Human Phenotype"

_cells, 2019, doi:10.3390/cells8010073_

Round 1
Reviewer 1 Report
Dear Editor,
the manuscript by Kara Turner et al. is on telomere biology , structure and function, as well as on telomere shortening in relation to human aging and age-related diseases
Few comments:
1) The relation between obesity and telomere shortening needs to be discussed more extensively.
It is known tha obesity associates with short telomeres due to inflammation.
It has also been shown that weight loss is associated to telomere lengthening in a positive correlation: the greater weight loss the greater telomere lengthening. ( Mol Genet Metab. 2016 Jun;118(2):138-42. Telomere length elongation after weight loss intervention in obese adults.Carulli ) could you please discuss it)
2) it has been shown that short telomeres and/or telomerase mutations associate with hepatocarcinoma. Please could you discuss also this point.
3)The manuscript is well written even though the references need to be updated.
Author Response
Dear Reviewer
Thank you for your helpful comments. We have incorporated your comments and responded them below.
Yours Sincerely
Dr Vimal Vasu (on behalf of all co-authors)
Response to Reviewer 1 Comments:
The relation between obesity and telomere shortening needs to be discussed more extensively. It is known that obesity associates with short telomeres due to inflammation. It has also been shown that weight loss is associated to telomere lengthening in a positive correlation: the greater weight loss the greater telomere lengthening (Mol Genet Metab. 2016 Jun;118(2):138-42. Telomere length elongation after weight loss intervention in obese adults.Carulli). Could you please discuss it?
Thank you very much for drawing our attention to this research publication, which we were not aware of. We chose to deliberately keep the detail on the relationship between telomere length and obesity brief in order to remain succinct and also to maintain flow within the review. Nonetheless, in light of this comment, we have updated table 1 to incorporate this detail in the section related to obesity. Please see the following modification to the text and the inclusion of the recommended reference (81):
“Telomeres are known to be shortened in obese individuals. Obesity is linked to a state of chronic inflammation, with increased reactive oxygen species (ROS) production in adipose tissue in addition to presence of greatly increased systemic oxidative stress. Furthermore, telomere length is correlated with body mass index (BMI), with increased BMI resulting in higher blood volume, stimulating increased proliferation of blood cells and leading to telomere shortening. Interestingly, weight loss is positively correlated with telomere lengthening, and those with shortest telomere length at baseline benefit from the most pronounced rate of telomere lengthening following weight loss. A greater adherence to a Mediterranean diet is also associated with longer telomeres.”
It has been shown that short telomeres and/or telomerase mutations associate with hepatocarcinoma. Please could you discuss also this point?
Many thanks for this comment. We have updated the text in lines 294-295 and included a further two references that the reader may be referred to for further information (162,163). Once again, we chose to be deliberately brief on the association between telomere length and specific types of cancer in the interest of remaining concise. We have not discussed any other specific cancer in greater detail and therefore we felt that further discussion around the relationship between telomere length and hepatocarcinoma would disrupt the flow of the review.
The manuscript is well written even though the references need to be updated.
We are very pleased to hear that you believe the manuscript to be well written. Thank you very much for this comment. We appreciate that many of the references in the bibliography are dated, however we chose to include original references that cite the first discoveries relating to telomere biology e.g. work by Elizabeth Blackburn, James Watson and Leonard Hayflick. We did not find any specific instruction with regards to publication date of references in the author instructions and therefore we have included references that cite notable observations during the history of the field in addition to the very latest observations. Indeed 64 of our references have been published in the last 10 years. We hope that this is satisfactory to the editor.
Reviewer 2 Report
In the review entitled «Telomere biology and human phenotype » the authors present the structure and function of human telomeres. Telomere homeostasis throughout a life-time is also addressed. Moreover the authors discuss the available evidence that shows that telomere shortening is related to human aging and the onset of age-related disease.
The review article is well written and it represents a significant contribution to the field. Nevertheless, I do have comments in order to improve the manuscript.
Line 21
“In all mammals, telomeres are formed of a highly conserved, non-coding, hexameric (TTAGGG) tandem repeat DNA sequence ……”
It is now know that mammalian telomeres are transcribed into telomeric repeat–containing RNA (TERRA). Azzalin et al 2007, as a consequence, I suggest to the authors to change the sentence line 21
Line 23
“ ….. and associated with specialised proteins collectively known as the Shelterin complex [1-3].”
To my point of view, the authors left out the non-Shelterin proteins present in the telosome. I suggest to the authors to modify the sentence in order to present all telomeric protein (Shelterin and the others) and to precise that they will focus only on the Shelterin in table1
Line 36
“Table 1. The Shelterin complex: Characterisation of the proteins that make up the telosome [3,6]. ”
Similar comment: Telosome is not only composed by Shelterin. Table1 title should be changed as only Shelterin proteins are listed. References “[3,6]” could be completed for i.e: Denchi and Sfeir molecular cell biology 2016
Table1 Protein name
Shelterin name should be modified according to their official name and the commonly use name mentioned
TRF1
Official full name telomeric repeat binding factor 1, TERF1, also known as TRF1
TRF2
Official full name telomeric repeat binding factor 2,TERF2, also known as TRF2
TIN2
Official full name TERF1 interacting nuclear factor 2, TINF2, also known as TIN2
POT1
Ok
TPP1
Official full name shelterin complex subunit and telomerase recruitment factor, ACD, also known as TPP1
RAP1
Official full name TERF2 interacting protein, TERF2IP, also known as RAP1
Table1 Interactions
TIN2 associates directly with TRF2 and TPP1 but not with POT1
Lines 62-64
“Thus, there are several mechanisms that result in telomere attrition following replication in proliferating cells and it is thought that the presence of a non-coding telomere sequence acts as a ‘buffering system’ to prevent the loss of crucial coding DNA”
To my point of view the notion of non-coding telomere sequence is not correct regarding TERRA presence. The sentence should be modified, references and comments should be added
Lines 83-84
“Despite the presence of these elongation mechanisms, telomere shortening is still observed following proliferation in most stem cells (with the exception of embryonic stem cells). This is because these mechanisms are increasingly reduced during differentiation and therefore they are insufficient to completely eradicate telomere loss.”
In these sentences maybe I do not understand the sentences, but to my point of view there is a problem between “in most stem cell” and “during differentiation”
Line 92
References [16,17], I will add Shay and Wright molecular cell biology 2000
Line 115
“….length in vivo in relation to chronological age.” I will add reference
Lines 115-116
“Moreover, a variety of studies have investigated the 115 relationship between telomere length and age-related disease.” I will add reference
Line 135
“…consistent with the highly variable telomere length observed in embryos [30] and fetuses [33,34].”
I will add human in the sentence “…. in human embryos [30] and fetuses [33,34].”
Line 166
I will modified the organization of this last part of the manuscript as follow:
4. Telomere length and biological aging
4.1.Telomere biology and premature aging disorders
4.2 Telomere length in age-related cardiometabolic and neurological disorders
4.3 Telomeres, tumorigenesis and cancer
Lines 168, 169, 180
in vitro, in vivo must be written in italics
Line 201
“…associated with type II diabetes” I will add reference
Line 255
“....[118,119]” I will add Chevret et al Blood 2014
Author Response
Dear Reviewer
Thank you for your helpful comments. We have incorporated your comments and responded them below.
Yours Sincerely
Dr Vimal Vasu (on behalf of all co-authors)
